# Flash Flood Detection via Copula-based IDF Curves: Evidence from Jamaica

Dino Collalti[1,2], Nekeisha Spencer[3], and Eric Strobl[1,2]

[1]Department of Economics, University of Bern
[2]Oeschger Centre for Climate Change Research, University of Bern
[3]Department of Economics, University of the West Indies

**Correspondence:** Dino Collalti (dino.collalti@windowslive.com)

**Abstract.**

Extreme rainfall events frequently cause hazardous floods in many parts of the world. With growing human exposure to floods, studying conditions that trigger floods is imperative. Flash floods, in particular, require well-defined models for the timely warning of the population at risk. Intensity-duration-frequency (IDF) curves are a common way to characterize rainfall and flood events. Here, the copula method is employed to model the dependence between the intensity and duration of rainfall events separately and flexibly from their respective marginal distribution. Information about the localization of 93 flash floods in Jamaica was gathered and linked to remote-sensing rainfall data and additional data on location-specific yearly maximum rainfall events was constructed. The estimated Normal copula has Weibull and generalized extreme value (GEV) marginals for duration and intensity, respectively. Due to the two samples, it is possible to pin down above which line in the intensity duration space a rainfall event likely triggers a flash flood. The parametric IDF curve with an associated return period of $2\frac{1}{6}$ years is determined as the optimal threshold for flash flood event classification. This methodology delivers a flexible approach to generating rainfall IDF curves that can directly be used to assess flash flood risk.

## 1   Introduction

Over the last twenty years, more people have been affected by floods than by any other natural disaster.[1] Among pluvial floods, flash floods have the highest average mortality (Jonkman, 2005; Hu et al., 2018). The Caribbean is especially at risk from flash floods. The region is particularly prone to hydro-meteorological hazards, urbanization is often unregulated, and soil degradation is common such that flash floods are frequent (Gencer, 2013; Pinos and Quesada-Román, 2021). For instance, heavy rain on March $5^{th}$ 2022 in Northern Hispaniola caused severe flash floods, leading to 2 deaths and hundreds being displaced.[2] Flash floods follow shortly after heavy rainfall and are highly localized phenomena that occur in basins of no more than a few hundred square kilometers and have a response time of a few hours (Amponsah et al., 2018). Steep slopes, impermeable surfaces, and

---

[1]Authors' calculation using EMDAT database. Since 2000, 1.7 Billion people have been affected by floods, followed by droughts (1.4 Billion), storms (0.8 Billion), and earthquakes (0.12 Billion).

[2]https://floodlist.com/ Accessed last on January $11^{th}$ 2023.

saturated soils are factors that can transform a heavy rainfall event into a flash flood hazard (Silvestro et al., 2019). The high localization and multidimensionality involved in flash floods make their study particularly involved.

It has long been a primary objective of weather service providers to create a warning system that connects rainfall to floods and landslides (Alfieri et al., 2012). Warning systems typically use some lower bound or threshold above which a warning would be issued (Hapuarachchi et al., 2011). Empirical thresholds for when rainfall events become hazardous connect the intensity ($I$) to the duration ($D$) and are used for the construction of so-called intensity-duration-frequency (IDF) curves (Koutsoyiannis et al., 1998). Commonly, estimation of IDF curves requires assumptions on the marginal distribution of $I$ and $D$ or the two marginals were assumed to be independent.[3] Using copula functions for conditional sampling allows the flexible and separate definition of marginals and dependence. Multiple studies employ the said method to estimate rainfall IDF curves for landslides and heavy rainfall events (Singh and Zhang, 2007; Ariff et al., 2012; Bezak et al., 2016; Li et al., 2019; Suresh and Pekkat, 2023). These studies often define the yearly maximum event of measurement stations by some decision rule and model the resulting time series. This allows for a good statistical fit and a well-described dependence between $I$ and $D$. However, since the data does not necessarily contain hazardous events, little inference can be made about these.

The calculation of rainfall IDF curves is strongly rooted in extreme value theory (EVT) since the intensity and duration of extreme events lend themselves as a relevant application (Koutsoyiannis et al., 1998). It is a common approach to flexibly derive IDF curves that can accommodate non-linearity in its parameters, e.g. to depict climatic time trends (Hosseinzadehtalaei et al., 2020; Sam et al., 2023). More recently, the use of copula functions to extend univariate EVT to multivariate extreme value theory (MVET) has been suggested as a complement to existing approaches, including IDF curves (Renard and Lang, 2007; Salvadori and De Michele, 2010; Chen et al., 2019). The main difference between the two approaches is the focus of analysis. Taking a multivariate perspective with copula functions, one focuses on the potentially non-linear dependence between intensity and duration (Bezak et al., 2016). In contrast, the univariate EVT approach typically focuses on univariate characteristics such as non-stationarity in distributional components (Martel et al., 2021). In principle, a multivariate approach with copula functions could also incorporate non-stationarity in an appropriate methodology. However, such a procedure becomes increasingly complex and requires sufficient data (Li et al., 2019).

This study aims to construct IDF curves with information from confirmed flash flood events in Jamaica. This allows for inference with regard to the hazard by comparing the odds ratio of flood occurrence given a frequency, where less frequent events are more severe and vice versa. The calculation of the odds ratio requires a set of extreme but non-hazardous events as well as a set of hazardous rainfall events. Following the literature, the local yearly maximum rainfall events are defined. Additionally, a complete and confirmed list of Jamaican flash floods by the Office of Disaster and Preparedness Management (ODPEM) is utilized to define hazardous events. These observed flash flood events are linked with 11 km × 11 km cells of remote sensing rainfall information. These remotely sensed data have several advantages compared to station data, such as consistency in sensors and resolution. While direct in-situ measurements are factual, they depend on the location and continuous operation of stations. Currently, the number of modern automatic weather stations in Jamaica is well below the remote sensing

---

[3]There are also some instances where a specific dependence has been assumed from theoretical considerations, see Koutsoyiannis et al. (1998)

resolution, with the exception of the area around the capital Kingston.[4] Subsequently, the IDF curve threshold, which separates
the confirmed hazard events from the rest via odds ratio, is determined. This threshold can serve as a simple decision rule for
the identification of flash flood triggering rainfall events.

There are a number of reasons why the Caribbean and Jamaica in particular is an interesting case study. Small island states
in the Caribbean have long been identified as especially vulnerable to extreme meteorological events and associated flooding
(IPCC, 2012; Wilson et al., 2014). Moreover, extreme precipitation events have shown an increase since 1950 in the Caribbean
region (Peterson et al., 2002; Stephenson et al., 2014). At the same time, there is little information on the local rainfall risk.
In this regard, it is common practice to transfer IDF curves for some Caribbean island nations to others, despite their different
rainfall characteristics (Lumbroso et al., 2011). Burgess et al. (2015) therefore developed IDF curves for Jamaica with long
historical data. Linearly projecting the historical parameter estimates to 2100, they find that the intensity of a 100-year return
event increases by 27% to 59% as a result of increasing variability due to climate change.

Quantifying extreme rainfall-induced hazards has important applications, such as for risk maps, warning systems, or re-
insurance schemes, particularly for the Caribbean. For example, the Climate Risk Early Warning Systems (CREWS) aims
to strengthen hydro-meteorological and early warning services in the Caribbean, focusing on hurricanes and other hydro-
meteorological hazards. Its first assessment in 2015 identified the need for increased forecasting of secondary hazards such
as coastal flooding and flash floods. Currently, pilot projects to strengthen national multi-hazard early warning systems in
the Caribbean community countries are devised through CREWS. The Caribbean Risk Information System (CRIS) platform,
created by the Caribbean Disaster Emergency Management Agency (CDEMA), aims to support informed decision-making by
providing access to information on hazards and does so via geospatial data for risk and hazard mapping, disaster preparedness,
and response operations. This input data relies on research in the hazard, exposure, and vulnerability domain.[5] Another example
is the Caribbean Catastrophe Risk Insurance Facility (CCRIF), which since 2013 has provided insurance against excess rainfall
to member countries (Linkin, 2014). More specifically, its CCRIF Excess Rainfall (XSR) product is a parametric insurance
based on specific rainfall thresholds that determine payouts.

The remainder of the paper is organized as follows: Section 2 presents the study region and describes the data. Section 3
details the methodology of conditional copula modeling and how the two samples are used to determine an IDF curve based
flash flood threshold. Section 4 then presents the results. Section 5 discusses the findings and section 6 concludes.

---

[4]http://metservice.gov.jm/aws/ Accessed last on January $3^{rd}$ 2023.

[5]Most commonly, risk is defined as the combination of the three components hazard, exposure, and vulnerability. Hazard relates to the physical phe-
nomenon, in this case, flash floods. Exposure could be in terms of people, buildings, or economic assets at risk of the hazard. Vulnerability then links the
hazard to the exposure and translates it to risk. For instance, given a flash flood hazard, the vulnerability of urban or agricultural settlements (exposure) is
different, and as such, the risk is different as well.

 ## 2  Study Region & Data

### 2.1  Study Region

Jamaica is the third-largest Caribbean island by land area after Cuba and Hispaniola. The island's topography is characterized by interior mountain ranges descending to coastal plains where the eastern Blue Mountains historically experience the most rainfall (Climate Studies Group Mona, 2020). Jamaica lies in the Atlantic Hurricane Belt and is especially at risk of climate change (Monioudi et al., 2018). Tropical cyclones and the accompanying heavy rainfall are frequent and cause severe destruction (Spencer and Polachek, 2015; Collalti and Strobl, 2022). For instance, between June 2007 and August 2021, the CCRIF made 54 payouts for a total of USD 245 million, of which USD 135 million are for tropical cyclones, USD 60 million for excess rainfall and USD 49 million for earthquakes (mainly the devastating 2021 Haiti earthquake). Thus, local susceptibility to floods has become vital to planning and development in Jamaica (Nandi et al., 2016).

### 2.2  Rainfall Climate

The annual cycle of rainfall for Jamaica reflects a bimodal pattern with rainfall peaks in May and October typical for the North-Western Caribbean (Martinez et al., 2019). This pattern is a result of the interplay between the large-scale climatic modulators of the region, namely the North Atlantic High pressure system ("Azore High"), the seasonal warming of the Atlantic, and the Atlantic Trade Winds (Climate Studies Group Mona, 2020). That is, the north-to-south movement of the North Atlantic High in boreal autumn (south-to-north movement in spring), the Atlantic cooling in autumn (warming in spring), coupled with the trade wind inversion in boreal winter cause boreal winter (and to a lesser degree summer) to be dry. Looking at the spatial rainfall distribution over Jamaica in Figure 1, one sees that the spatial distribution of rainfall depends on the weather system at different temporal scales. On average, during the study period from 2000 to 2019, most precipitation falls in the central north-western area of the Island and, to a lesser degree, on the north-eastern shore close to the Blue Mountains. Comparing this to the average total precipitation of events with a 6-hour, 24-hour, and 72-hour inter-event time definition (IETD), we can conclude that the Western part of the Island is subject to the most rainfall-heavy events with long durations whereas the Eastern part experiences the heaviest short duration events.

### 2.3  Flash Floods

The source of flash flood information is the Office of Disaster and Preparedness Management (ODPEM), whose responsibility includes monitoring extreme weather events in Jamaica and implementing measures to mitigate their impact. From the ODPEM, shapefiles of all 48 known flood events are obtained from 2001 to 2018. Many of these events correspond to a specific meteorological event, like a tropical storm that caused flooding in more than one location in Jamaica. We treat each event location separately if it falls uniquely in a remote-sensing rainfall cell. For example, during heavy rain on May $14th-15th$ in 2017, several places around Cave Valley (parish of St. Ann) in central Jamaica, as well as, to the south, around Morgan's Pass in the parish of Clarendon, experienced severe flooding. These locations are approximately 20 km apart, lie on two sides

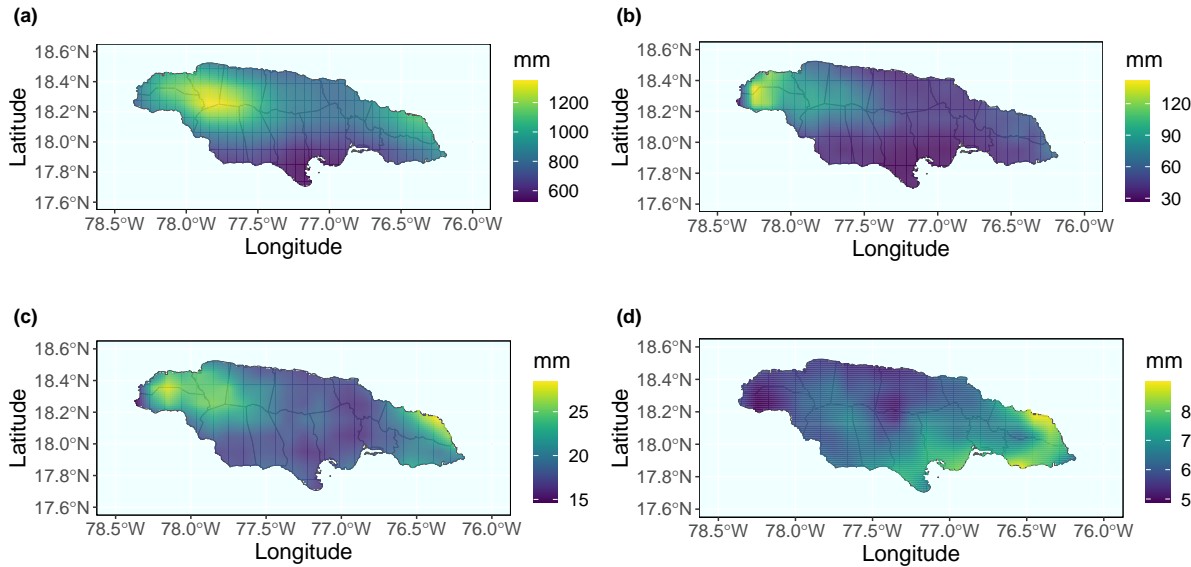

**Figure 1.** Maps of the (bilinearly interpolated) rainfall climate in Jamaica from the precipitation data as used in this study. a) shows the average yearly precipitation, b) the average precipitation of events with a 72-hour IETD, c) the average precipitation of events with a 24-hour IETD, and d) the average precipitation of events with a 6-hour IETD.

of the north/south watershed, and are thus treated as two incidents in their respective rainfall cell. Some flood events in the OPDEM shapefiles could not be verified by any report and were therefore dropped, as were a few riverine floods that would require explicit hydrological modeling, which is beyond the scope of this study. Some events where the exact day(s) are not included in the data are identified using local newspaper reports. A total of 93 flash flood events were localized for Jamaica
with approximate timing.

## 2.4   Precipitation

The source for precipitation data is Version 06B of the Global Precipitation Measurement (GPM) Integrated Multi-satellitE Retrievals (IMERG, Huffman et al. (2015)). The satellite precipitation algorithm combines microwave and infrared precipitation measurements to produce precipitation estimates, adjusted with surface gauge data. The resulting product is a half-hourly
data set with near-global coverage at a $0.1° \times 0.1°$ resolution since June 2000. Compared to other remote sensing or reanalysis products, the GPM-IMERG has a considerably higher spatial and temporal resolution than its competitors. Also, the number of distinct cells and, thus, spatial resolution is considerably higher than the number of measurement stations in Jamaica. One major drawback of the GPM-IMERG, its short timeframe, does not apply to this study because all the OPDEM events are fully captured in the observational period since June 2000. Note that the quality of satellite rainfall data has leapfrogged in
the last decade: An inter-comparison of rain-gauge, radar, and GPM-IMERG for rainfall-runoff modeling by Gilewski and

Nawalany (2018) in a mountainous catchment in Poland identified that radar and GPM-IMERG outperform rain-gauge data. Tang et al. (2020) provides a comprehensive overview of different satellite precipitation and reanalysis products, reporting good performance for GPM-IMERG and it is continuously improving in more recent versions.

## 3 Methodology

In this methodology section, we describe how to create IDF curves from the GPM-IMERG rainfall data via copula functions. Furthermore, we present a flash flood classification of extreme rainfall events by connecting the IDF curves with confirmed flash flood events. The methodology is summarized in Figure 2. Core to the methodology are the event definition, copula selection and estimation, marginal distribution selection and estimation, conditional copula sampling, and the classification via odds ratio. Subsequently, these methods are explained and their application is presented in the results Section 4.

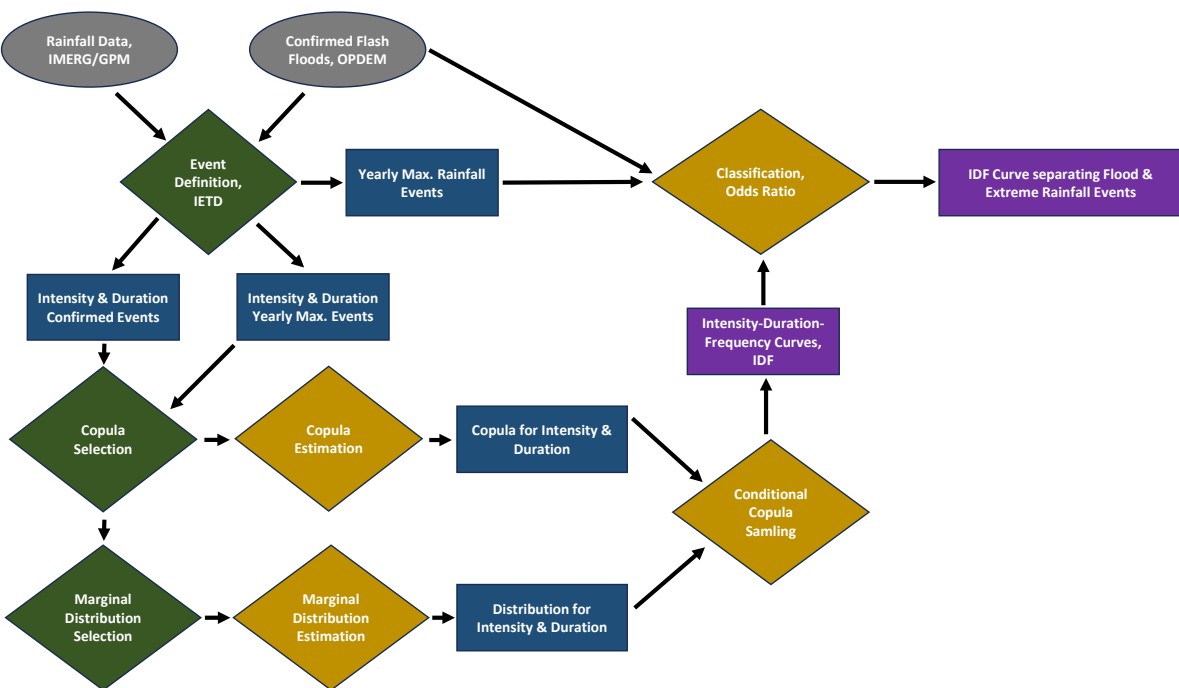

**Figure 2.** Flowchart illustrating the methodology. In grey are the original data inputs. Diamond squares are methods and procedures applied, whereas regular squares are outcomes of said methods and procedures. In green are methods that involve choices outside regular statistical testing such that they do not contribute to the analysis of uncertainty in Section 4.6 whereas methods in gold do. Blue squares are intermediary outcomes whereas purple are final outcomes that are subject to discussion in Section 5.

## 3.1 Event Definition

The data on confirmed flash flood events provides location and start date information, but no sub-daily timing of rainfall onset and its ending. We thus need to find and define rainfall events that start before the flood (potentially lasting longer than the reported date). We resort to the common inter-event time definition (IETD) method to delimit the events (Ariff et al., 2012; Bezak et al., 2016). The IETD refers to the minimum duration without rain between consecutive rainfall events. An IETD of a few hours is typically selected for floods, while for landslides the IETD is longer, i.e., up to several days (Melillo et al., 2015). For confirmed events, the event definition starts with a window of +/- 7 days around the date given by the OPDEM or newspapers. Within that window, the event with the maximum cumulative rainfall is regarded as the flood-inducing rainfall event. Figure 3 illustrates the procedure. The yearly maximum events are constructed the same way, though for each cell each year is considered separately. Note that a minimum threshold of $0.1 \text{ mm h}^{-1}$ for a given observation to start an event is imposed to reduce the number of events.

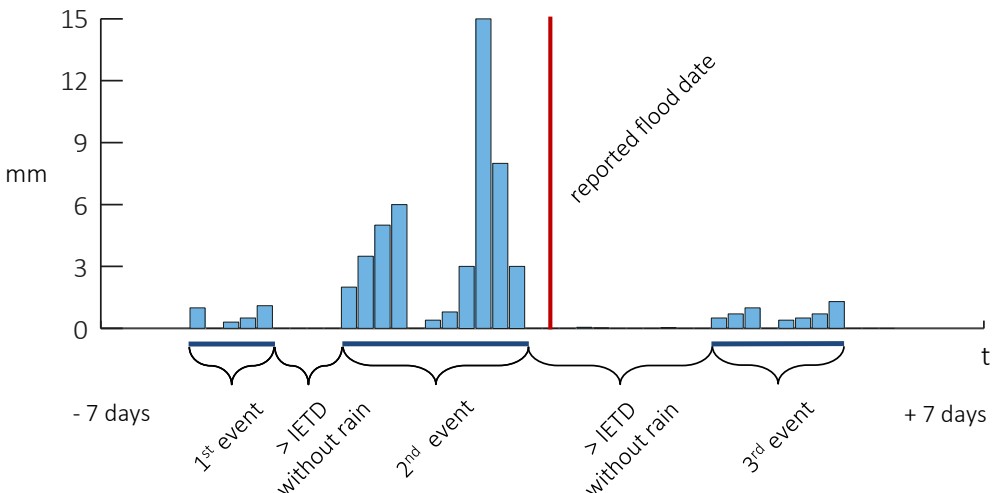

**Figure 3.** Illustration of the event definition in the case of a confirmed event with the reported date from OPDEM or newspapers in red. In the time frame +/- 7 days around this date, three separate events are defined given an IETD and a minimum threshold of $0.1 \text{ mm h}^{-1}$. Three events result, where the second event is the maximum event measured by cumulative rainfall and is considered the flood-inducing rainfall event.

## 3.2 Conditional Copula Modelling

Informally, copulas can be described as "functions that join or couple multivariate distribution functions to their one-dimensional marginal distribution functions" (Nelsen, 2007). More formally, given a 2-dimensional (joint) distribution function $H$ with uni-

variate margins $F_1 and F_2$, there exists, by the first part of Sklar's Theorem, a 2-dimensional copula $C$ such that

$$H(\boldsymbol{x}) = C\big(F_1(x_1), F_2(x_2)\big), \quad \boldsymbol{x} \in \mathbb{R}^2. \tag{1}$$

The copula $C$ is uniquely defined on $\prod_{j=j}^{2} \mathrm{ran} F_j$ and there given by

$$C(\boldsymbol{u}) = H\big(F_1^{\leftarrow}(u_1), F_2^{\leftarrow}(u_2)\big), \quad \boldsymbol{u} \in \prod_{j=1}^{d} \mathrm{ran}\, F_j, \tag{2}$$

where $F^{\leftarrow}$ denotes the generalized inverse, which equals the regular inverse $F^{-1}$ for continuous and strictly increasing distribution functions (dfs). By the definition of a cumulative distribution function, $\mathrm{ran}\, F_j \in (0,1)$ such that the copulas univariate margins are standard uniform $U(0,1)$ (Hofert et al., 2018). Three attributes follow: the copula function (1) uniquely specifies the dependence for the whole distribution, (2) can be recovered from data on joint and marginal distribution, and (3) imposes no constraint on the shape of the dependence.

The conditional copula method has previously been used to estimate IDF curves (Singh and Zhang, 2007; Ariff et al., 2012; Bezak et al., 2016; Li et al., 2019). Let $C$ be the 2-dimensional copula and let $\boldsymbol{U} \sim C$, $u_1 \in (0,1)$ and $u_2 \in [0,1]$, then

$$C_{2|1}(u_2|u_1) = \mathbb{P}(U_j \le u_j | U_1 = u_1). \tag{3}$$

If one fixes for some value of $u_1 \in (0,1)$, the conditional copula function $C_{2|1}(u_2|u_1)$ is a distribution function on $[0,1]$ and can be used for conditional sampling. The evaluation of $C_{2|1}(u_2|u_1)$ however involves the evaluation of partial derivatives instead of densities (Hofert et al., 2012, 2018). The theoretical basis for the process is the inverse Rosenblatt transformation, also known as the conditional distribution method.

Consider that $C_{U,V}(u,v)$ is the copula function of interest and let intensity $I = i$ and duration $D = d$ have marginal distribution functions $V = F_I(i)$ and $U = F_D(d)$. For a known value of $U = u$, $C_{V|U=u}$ gives realizations of marginal $V$. The corresponding value of $u$ can be obtained by the marginal distribution function. From $u$ and $v$, the respective $i$ and $d$ can be recovered easily since $d = F_D^{-1}(u)$ and $i = F_I^{-1}(v)$. The conditional copula function can be written as

$$C_{V|U=u}(v|U=u) = \frac{\partial}{\partial u} C_{U,V}(u,v)\Big|_{U=u.} \tag{4}$$

The conditional copula, which is a conditional bivariate distribution, relates to the return period $T$ as follows

$$C_{V|U=v}(v|U=u) = 1 - \frac{1}{T}. \tag{5}$$

For a given value of $u$ and a return period $T$, solving Equation 4 and 5 simultaneously yields the corresponding $v$. Via the marginal distribution function, the respective values of $i$ and $d$ are recovered and represent a point on the IDF curve for a return period $T$. For every return period $T$, many values of $u$ are chosen to get an approximately smooth IDF curve. That process is repeated for other $T$ to construct IDF curves which are increasing in severity with $T$.

### 3.3 Two Sample Approach

Rainfall events of interest are those that lead to flash floods. However, a block maxima approach, partitioning the data into yearly blocks, allows a direct relation with return periods and is thus often chosen (Ariff et al., 2012). The proposed methodology uses information from block maxima as well as confirmed flood events. There are $m = 93$ confirmed flood events and, at these locations, $n = 1120$ yearly cell-wise maxima. The yearly maximum events serve to estimate the copula function and the marginal distributions of intensity and duration for these extreme rainfall events. Conditional sampling from the copula enables the construction of IDF curves with $T$ year return periods. One can then derive the IDF curve associated with a certain return period above which the likelihood of flash flood occurrence is maximized. For every return period the IDF curve is recovered and the ratio $R$ of confirmed flash flood events $m$ against the number of yearly maximum rainfall events $n$ that lies above that curve is calculated,

$$R = \frac{\sum_{i=1}^{m} I\big(d_i \geq (\tilde{d}|U = u_i)\big)}{\sum_{j=1}^{n} I\big(d_j \geq (\tilde{d}|U = u_j)\big)}, \tag{6}$$

where $(\tilde{d}|U = u_i)$ is the estimated duration via conditional copula sampling and marginal transformation $\tilde{d} = F_D^{-1}(\tilde{u})$. The IDF curve with a return period associated with the highest ratio $R_{r:max}$ is the one that separates the events from non-events best. This constitutes a so-called critical layer $(d, i) \in L^2 : 1 - H((d, i) = P(D > d, I > i) = t$ where all combinations of $i$ and $d \in L^2$ have the same probability $1 - H((d, i) = t$ (Salvadori et al., 2016). The critical region, which corresponds to a flash flood classification, is defined as $L_t^> = \{(i, d) \in L^2 : 1 - H(i, d) < t\}$ (De Michele et al., 2013). Subsequently, the return period $T^>$ of an event in the critical region is defined by the inverse probability of falling into the critical region (Zscheischler et al., 2017):

$$T^> = \frac{\mu}{P((D, I) \in L_t^>)}, \tag{7}$$

where $\mu$ denotes the average time unit, which is 1 year in the case of yearly maxima.

### 3.4 Candidate Copulas

The selection of appropriate copula is carried out in two steps. First, a set of candidate copulas is defined. Second, the candidate copula is compared on the basis of fit, for both the event and the yearly maxima data. The first restriction on candidate copulas is that a conditional sampling algorithm exists. This is the case for the families of Archimedean and elliptical copulas (Hofert et al., 2018). The literature on landslides and flash flood IDF curves has further established the negative relation between an event's duration and its intensity, which is the second restriction on candidates (Aleotti, 2004; Piciullo et al., 2017). Table 1 shows the copulas for which conditional sampling algorithms exist and some of their properties. The two restrictions leave one with three potential copula classes, Normal, Frank, and $t$ copula. Note that these copulas are all radially symmetric and exchangeable. Geometrically, radial symmetry is the symmetry of the density with respect to the point $\mathbf{1/2} = (1/2, ..., 1/2)$. Exchangeability is the symmetry of the density with respect to the main diagonal. Given a negative dependence, a copula that is not radial symmetric is one whose lower tail dependence is different from its upper tail dependence, whereas a copula that

**Table 1.** Candidate Copula Families

| Name | Attainable Dependence | Radial-Symmetry | Exchange-ability | Negative Dependence |
|---|---|---|---|---|
| Gaussian | (-1,1) | ✓ | ✓ | ✓ |
| $t_v$ | (-1,1) | ✓ | ✓ | ✓ |
| AMH | [0, 1/3) | | ✓ | |
| C | [0, 1) | | ✓ | |
| F | (-1, 1) | ✓ | ✓ | ✓ |
| GH | [0, 1) | ✓ | ✓ | |
| J | [0, 1) | ✓ | ✓ | |

Overview of candidate copula families with respect to their attributes.

is not exchangeable is one whose dependence changes with the order of the marginals. The best candidate copula is selected on the basis of the Cross-Validation Copula Criterion (CIC) by Grønneberg and Hjort (2014), which is an Akaike Information Criterion (AIC)-like criterion on a Maximum-Pseudo-Likelihood Estimate (MPLE) of semi-parametrically (i.e., with non-parametric estimated margins) estimated copula. The methodology is implemented in the R package "Copula", with which all the subsequent copula modeling is carried out (Kojadinovic and Yan, 2010; Hofert et al., 2018).

### 3.5 Selection of Marginals

The IDF curve construction via conditional copula in section 3.2 requires the estimation of marginals for duration $D$ and intensity $I$. Candidate marginal distributions are the Weibull, Gamma, Log-normal, and Generalized-Extreme-Value (GEV) distributions, and are all estimated via maximum likelihood and assessed by AIC. The empirical probability density function is also inspected graphically against the marginal distributions' estimated probability density to assert its fit.

## 4 Results

### 4.1 Event Definition

The event definition in time requires an appropriate IETD. Values of IETD between 4 h and 24 h are considered. Graphical assessment of mean event intensity and duration revealed that an IETD of 12 hours best delimits the rainfall events. See Appendix section A1 for a discussion and graphical examples for various IETD. Figure 4 shows the locations of confirmed flash flood events in Jamaica and the average intensity, duration, and total rainfall of the yearly maximum events. Average intensity is highest inland and to the West, but fairly evenly distributed. Average total rainfall is highest at the eastern shore north of the Blue Mountains, with a second agglomeration of high total rainfall cells in the West. Duration exhibits a similar pattern to total rainfall, with the longest events in the West. Since the confirmed flash flood events are evenly distributed across

the island, variation in local conditions is expected to be captured well. Table A1 in the Appendix further provides summary statistics for both yearly maximum and confirmed flash flood events.

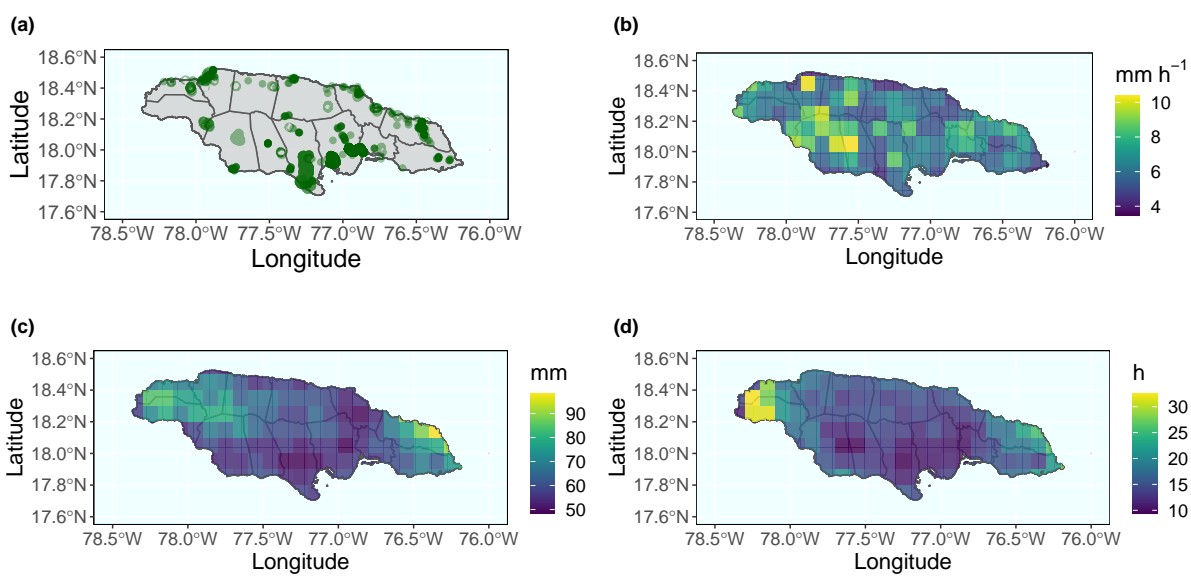

**Figure 4.** (a) Location of the confirmed flash flood events, (b) cell-wise mean intensity of locations maximum events, (c) cell-wise mean total rainfall of locations maximum events, and (d) cell-wise mean duration of locations maximum events.

## 4.2 Copula Selection

The shape of dependence can be assessed via pseudo-observations. Pseudo-observations are obtained by first estimating the empirical distribution functions $F_n(n, j)$ for $j \in (I, D)$,

$$F_{n,j} = \frac{1}{n+1} \sum_{i=1}^{n} 1(X_{i,j} < x), \quad x \in \mathbb{R}, \tag{8}$$

where $1(\cdot)$ is the indicator function. These estimated margins can then be used to form the sample:

$$U_{i,n} = \big( F_{n,D}(X_{i,D}), F_{n,I}(X_{i,I}) \big), \quad i \in \{1, ..., n\}. \tag{9}$$

Figure 5 displays the pseudo-observations and demonstrates a strong negative dependence in both samples. This limits the set of potential copulas to the Normal, Frank, and $t$ copula. Note that these copulas are all radially symmetric and exchangeable.

Estimates of the copula information criterion (CIC) are shown in Table 2. Selecting the Frank copula for the yearly maximum events leads to a higher CIC than the Normal or $t$ copula. For the confirmed events, selecting the Frank copula leads to a lower CIC than the Normal or $t$ copula.

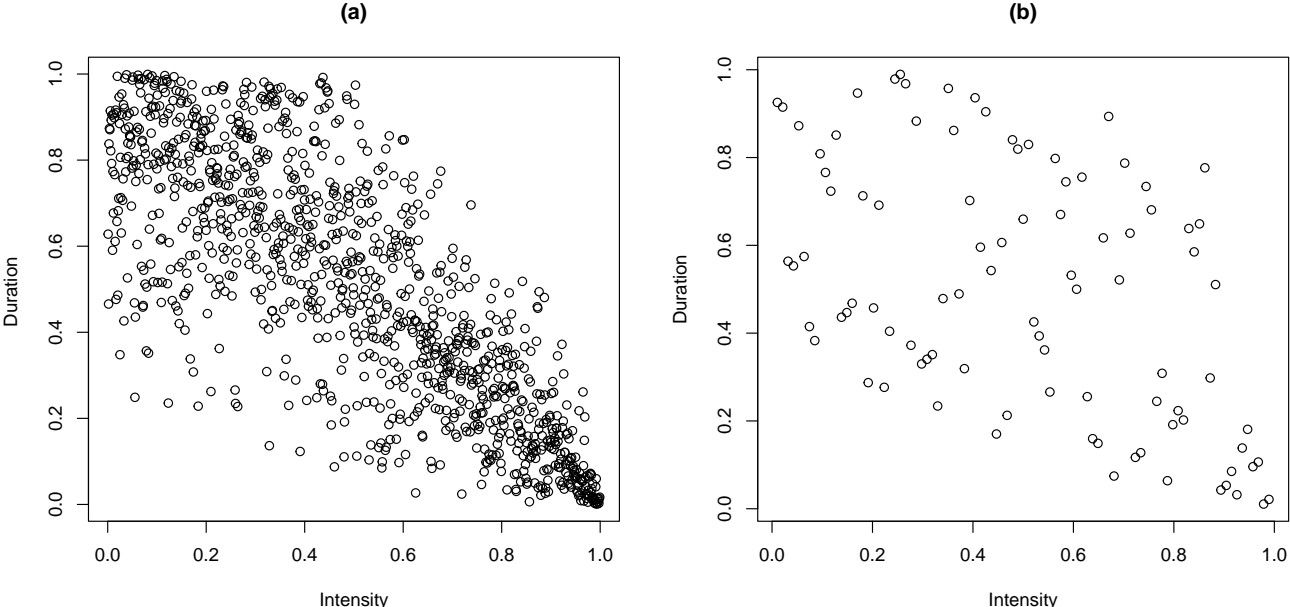

**Figure 5.** (a) Pseudo-observation of the yearly maximum rainfall events (YME) and (b) of the confirmed flash flood events (FFE). Ties in the duration variable due to the measurement scale are randomly split.

**Table 2.** Copula Cross-Validation Criterion

|  | Normal Copula | Frank Copula | $t$ Copula |
|---|---|---|---|
| Maximum Events | 513.3 | 551.4 | 521.3 |
| Confirmed Events | 15.92 | 12.02 | 15.93 |

Cross-Validation Copula Criterion (CIC) by Grønneberg and Hjort (2014) for both data samples. The $t$ copula assumes $10 = v$ degrees of freedom.

Figure 6 displays pseudo-observations for both samples as well as a random sample of pseudo-observation under Frank and Normal copula. The sample of confirmed flash floods is too small to draw conclusive evidence regarding the optimal copula. There is also no clear visual indication around the locus of points for the Frank copula over the Normal copula or vice versa. However, the Normal copula exhibits tail dependence similar to the data, while the Frank copula is tail quadrant independent (Joe, 2014).

In summary, the Normal copula is better suited for the data and thus chosen for the analysis. It is more appropriate for yearly maximum events, which are the data on which the IDF curves are generated, as outlined in section 3.3 concerning the two sample approach. Additionally, the Normal copula is suitable for the confirmed events, as the CIC and graphical evidence shows.

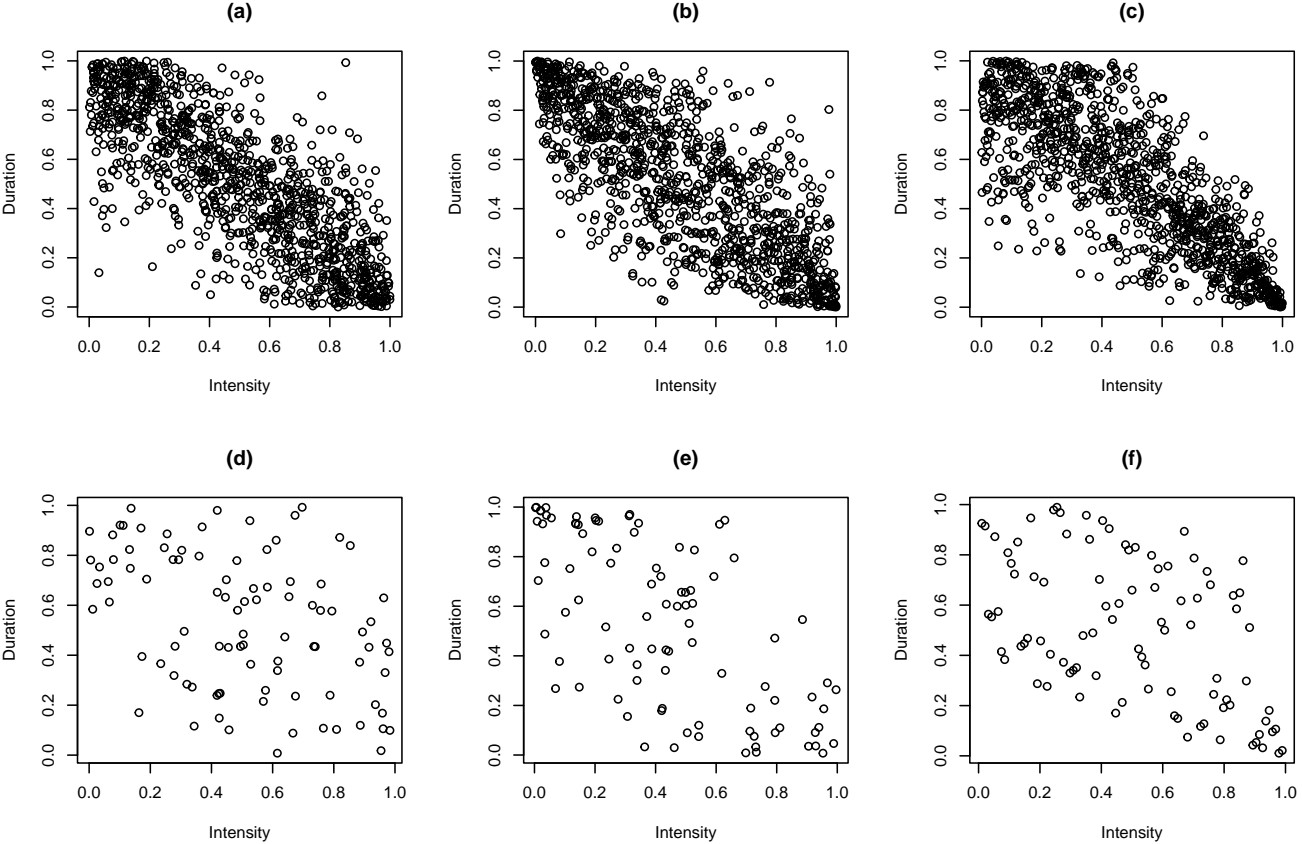

**Figure 6.** (a) random sample ($n = 1120$) of pseudo-observation of the yearly maximum rainfall events (YME) under the assumption of Frank copula, (b) under the assumption of Normal copula, and (c) the true pseudo-observation. (d) Random sample ($n = 93$) of pseudo-observation of the confirmed flash flood events (FFE) under the assumption of Frank copula, (e) under the assumption of Normal copula, and (f) the true pseudo-observation.

### 4.3 Estimation of Marginals

Table 3 reports the Akaike Information Criterion for both samples, where the parametric distributions are fitted via MLE to
250 the marginals. For both samples' intensity, the generalized extreme-value (GEV) distribution results in the lowest information loss. Similarly, for the duration, the Weibull distribution yields, in both instances, the lowest AIC.

Figure 7 compares the estimated distributions and the empirical probability density. In all cases the two match well. The confirmed flood events are on average slightly longer (11.4 hours versus 9 hours) and less intense (5.2 mm h$^{-1}$ versus 9.9 mm h$^{-1}$) compared to the yearly maximum events. Notably, the flood events are not as smoothly distributed due to the smaller
sample size. Both samples yield similar distributions and agree on the shape. Subsequent conditional copula modeling focuses on the more precisely estimated distributions from the large sample of yearly maximum events.

**Table 3.** Akaike Information Criterion

| | Yearly Maximum Events | | | |
|---|---|---|---|---|
| | Weibull | Gamma | Log-Normal | GEV |
| Intensity | 7374.2 | 7327.5 | 7059.3 | 7029.1 |
| Duration | 7053.8 | 7061.3 | 7181.4 | 7196.5 |
| | Flash Flood Events | | | |
| | Weibull | Gamma | Log-Normal | GEV |
| Intensity | 493.7 | 487.8 | 447.5 | 428.8 |
| Duration | 632.7 | 634.6 | 650.2 | 648.3 |

Akaike Information Criterion (AIC) for both samples. A parametric
distribution was fitted via MLE to the marginals, intensity, and duration.

## 4.4 Conditional Copula IDF Curves

Given the conditional copula modeling presented in Section 3.2 and the estimates for the GEV distribution of intensity, the Weibull distribution of duration, and the Normal copula for dependence, the following procedure can be employed to same from a conditional copula. Let's create a vector of quantiles $u \in [0,1]$ and set a return period $T$ relating to the conditional copula as in Equation 5, $C_{V|U=u}(v|U=u) = 1 - \frac{1}{T}$ where $C(\cdot)$ is the Normal copula function estimated in Section 4.2. Then for every quantile of $u$ and for every return period, the corresponding (conditional) quantile of $v$ results, thus giving a triplet of $(u_q, v_q, T)$ for any quantile $q$ of $u$ and return period $T$. Via the inverse distribution transformations $d = F_D^{-1}(u_q)$ and $i = F_I^{-1}(v_q)$, and the estimated marginal distributions from Section 4.3, that triplet becomes $(d, i, T)$. Connecting all points in the $(d, i)$ space for some $T$ then gives us an IDF curve for return period $T$ from conditional copula modeling.

IDF curves corresponding to return periods between 2 and 40 years are shown in Figure 8. The curves are all convex, such that shorter events have a disproportionately higher intensity. Visually, the choice of marginals has little impact on the IDF curves. Higher return periods shift the IDF curve outwards to higher intensities for all durations. Interestingly, convexity decreases with higher return periods. Even though longer return periods might be of interest, we are cautious that long return periods might not be appropriate given the available data. Coles et al. (2001) for instance argues that increasing extrapolation to upper quantiles of the distribution and longer return periods is increasingly at risk of unverifiable assumptions and uncertainty. Therefore, the maximum return period here of 40 years is twice the length of the data used in model estimation, which is more than sufficient for the extreme rainfall event classification.

## 4.5 Best IDF Curve

IDF curves generated with the Normal copula, a generalized extreme-value distribution for intensity and a Weibull distribution for duration reliably quantify the joint severity of an event by linking a return period to it. The next step is to find the IDF curve

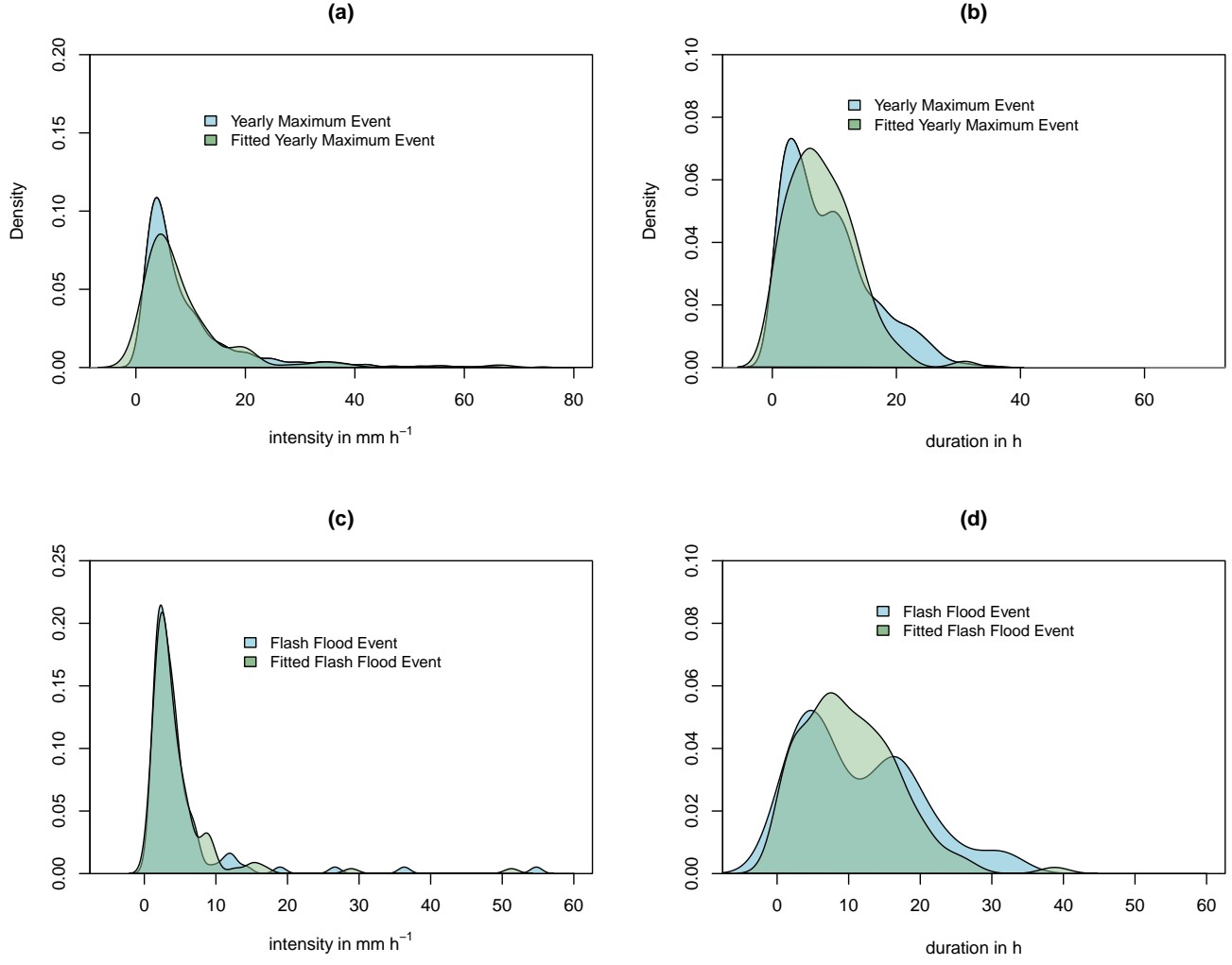

**Figure 7.** Comparison of estimated probability density function for (a) intensity of yearly maximum events (YME), (b) duration of yearly maximum events, (c) intensity of flash flood events, and d) duration of flash flood events (FFE).

above which the probability of a flood event is maximized. The highest odds ratio (0.66) is reached with a return period of 2 years and 2 months. Rainfall events that potentially trigger flash floods are thus expected to be at least as severe as a 2.17-year return period event.[6]

[6]This threshold is naturally higher than the simple empirical analog of the 93 confirmed events in Jamaica during the 18-year period because the geographical resolution is higher: looking at smaller scale areas, each of these areas' flood probability has to be lower than that of the whole island.

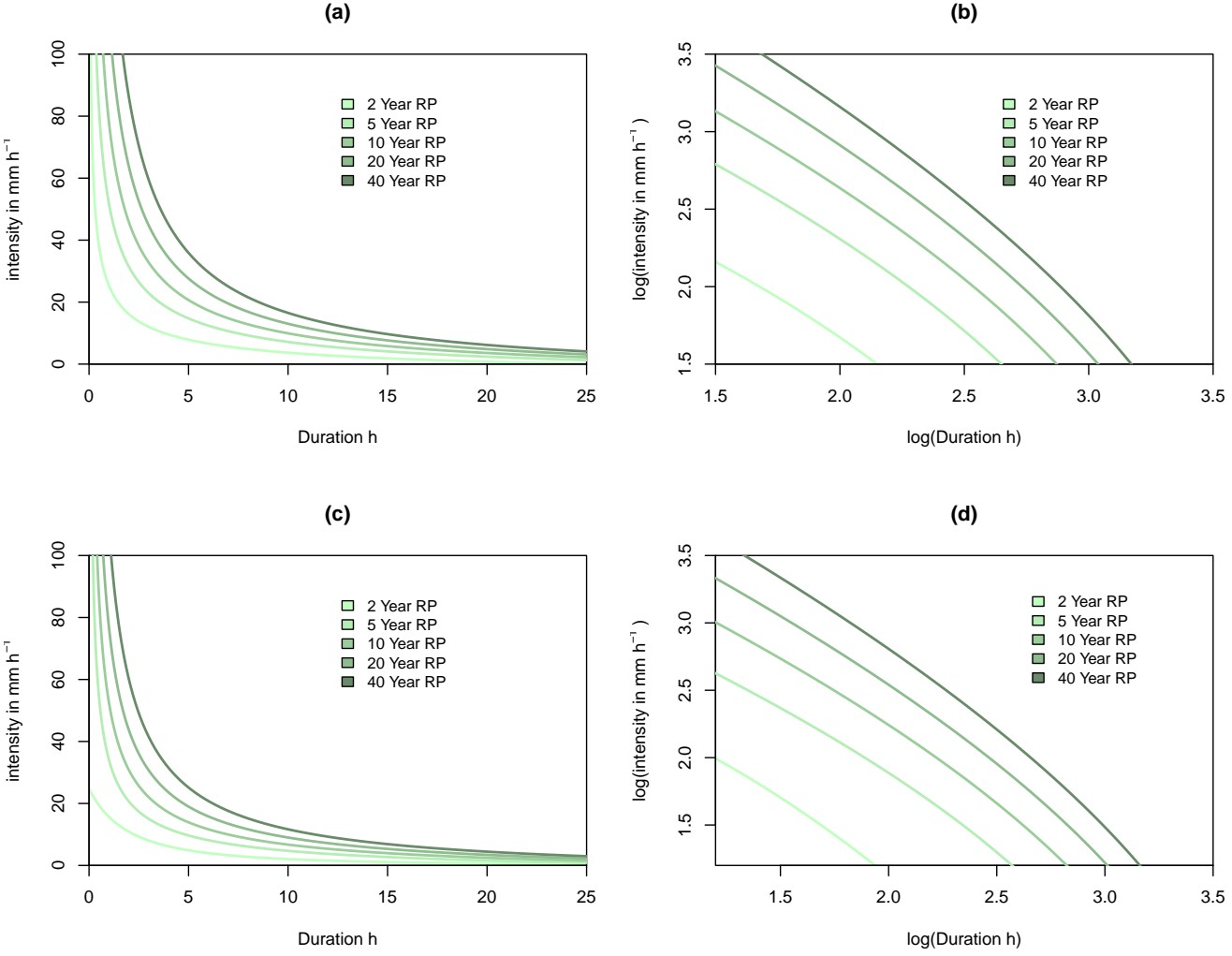

**Figure 8.** Intensity-Duration curves for frequencies corresponding to a return period of 2, 5, 10, 20, and 40 years. (a) shows these IDF curves for the Normal copula and marginals estimated from the yearly maximum events (YME), in logs in (b), and (c) shows the Normal copula from YME and marginals from flash flood events (FFE), in logs in (d).

## 4.6 Uncertainty

Each method and procedure in the methodology comes with uncertainty. In the case of the green-labeled procedure in Figure 2, this uncertainty is in terms of modeling choice, i.e. which approach to event definition or functional form of copula and marginal distribution is appropriate. Uncertainty from such modeling choices is difficult to quantify since there is no clear framework to define appropriate alternative hypotheses. In other words, the functions that have to be evaluated are not nec-

285 essarily nested models that can compared directly with a statistical test. In contrast, the gold-labeled methods in Figure 2 by design give measures of uncertainty or are nested to this uncertainty. Therefore, the uncertainty of the copula estimation, marginal distribution estimation, conditional copula sampling as well as odds ratio classification can be assessed, for instance with bootstrapping.

Bootstrapping is a method commonly used to create confidence intervals for complex problems, related to other resampling 290 methods such as cross-validation (Hesterberg, 2011). The approach has been employed to quantify the uncertainty of IDF curves (Sane et al., 2018). In our case, we use the standard bootstrap with replacement but randomly draw a sample of both the confirmed events and the yearly maximum rainfall events from the original data. We repeat that process many times and calculate the values of interest for the IDF curve and the ratio classification each time. This yields a distribution of these that mimics the real sampling procedure in the data. We repeat that process $1'000$ times and calculate confidence intervals as the 295 bootstrapped quantiles of the quantities of interest.

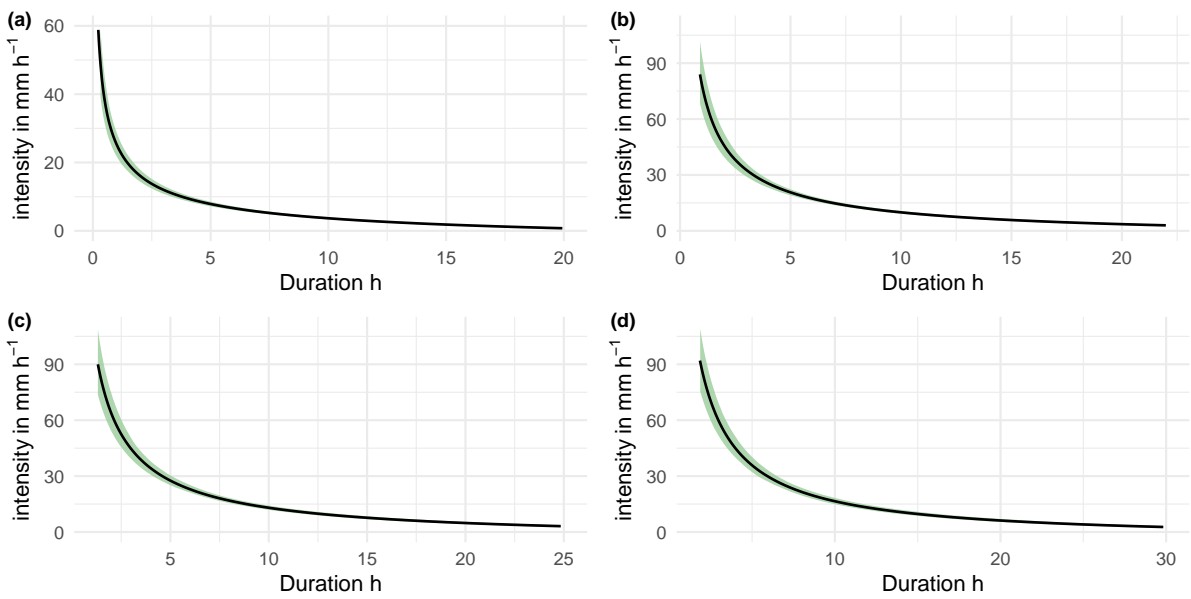

**Figure 9.** IDF curves with a return period of a) 2 years, b) 10 years, c) 20 years, and d) 40 years with a $99\%$-confidence band from bootstrapping over the estimation of marginals and the copula function.

Figure 9 shows IDF curves for return periods of 2, 10, 20, and 40 years with a $99\%$-confidence band from bootstrapping. There is very little uncertainty coming from the estimation of distribution and copula. We suspect that there could be more uncertainty from the modeling choices, albeit these choices are herein not discretionary but firmly grounded in statistical theory. Figure 10 shows the uncertainty of the return period classification ratio and the classification of yearly maximum rainfall events 300 with the optimal IDF curve. The uncertainty is much larger when classifying the events compared to the IDF curves. Not only is the uncertainty from the estimation of distribution and copula relevant, but many events are close to the optimal IDF curve

itself and thus within the range of estimation uncertainty. The sampling of confirmed and maximum rainfall events also directly affects the classification ratio by changing the composition of events that are evaluated.

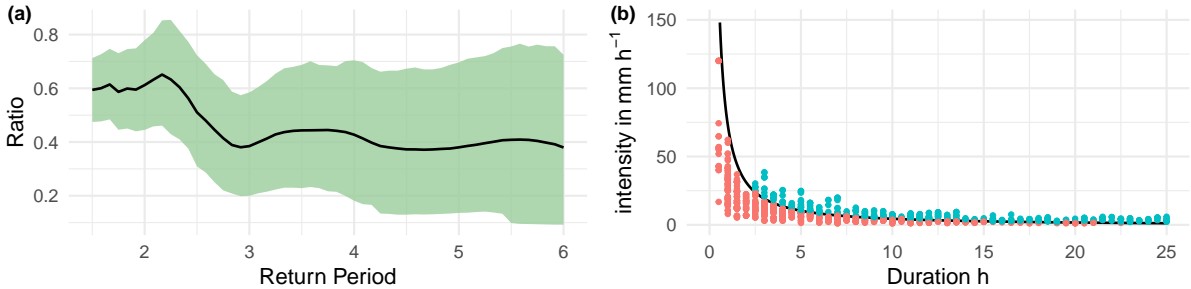

**Figure 10.** a) shows the ratio $R$ of confirmed flood events versus yearly maximum rainfall events for different return periods with a 95%-confidence band from bootstrapping over the estimation of marginals and the copula function. b) shows the IDF curve with a return period of 2.17 years and how this separates the yearly maximum rainfall events.

## 5 Discussion

It is insightful to compare the IDF curves from this study with those otherwise obtained for Jamaica. Burgess et al. (2015) provides Jamaica's most recent IDF curves, using long-time series data from two stations in Jamaica, extending existing annual maximum records back to 1895. With a return period of 5 years and a duration of 12 h, they estimate intensities of around 7.2 - 11.4 mm h$^{-1}$, depending on the configuration. For a duration of 2 h and again a return period of 5 years, intensities are between 32 - 33 mm h$^{-1}$. For a return period of 5 years, the results from the current study suggest an intensity of 7.14 mm h$^{-1}$ for 10 h and 22.8 mm h$^{-1}$ for 3 h. The corresponding IDF curves are thus in a similar range but are more strongly convex than those in Burgess et al. (2015), where dependence is not explicitly addressed. This might be caused by the choice of the Normal copula, which is well suited to depict convex dependence. It might also be caused by the type of data input in that Burgess et al. (2015) uses data from stations in the two largest cities in Jamaica, namely Kingston and Montego Bay, spanning back to 1895. In contrast, the remote sensing data employed in the current study covers the whole island, but only since 2000. Given the large difference in the investigated time period, climatic factors likely impact the results. Considering the spatial rainfall climate in Figure 1, there is a strong variation across the islands for different meteorological scales that changes with the duration of events in the satellite data. It is arguably advantageous for some applications to represent the island on average instead of in two specific locations, whereas the longer time series in (Burgess et al., 2015) are preferred in other applications.

Another kind of comparison can be made concerning the marginal distributions and studies that employ extreme value modeling for flood risk prediction. The intensity of confirmed flash flood events and extreme rainfall events in Jamaica is GEV-distributed. The shape parameter around $0.59$ implies that intensity is Fréchet extreme value distributed and has a lower limit. The duration of confirmed flash flood events and extreme rainfall events in Jamaica is Weibull distributed and has an upper limit. The lower limit of intensity and upper limit of duration are consistent with the IETD definition which forces events to

be relatively short and having a minimum intensity to be considered. Several studies also find evidence for Fréchet distributed
intensity when constructing IDF curves with extreme value analysis, though the shape of the distribution appears to be highly
case-sensitive (Sane et al., 2018; Bonaccorso et al., 2020; Yeo et al., 2021)

The quantification of extreme rainfall hazards through the IDF curve classification has direct applications for policymakers.
One may first consider the case of the CCRIF XSR parametric insurance against excess rainfall that is based on specific rainfall
thresholds for payouts. The most recent version, XSR 2.5, utilizes separate exposure, vulnerability, and hazard modules for
each member. For the Caribbean, the module is triggered by rainfall events that exceed some country-specific average intensity
threshold for 12 h (short events) or 48 h (long events). These country-specific thresholds are optimized to increase the likelihood
of detecting severe events while not capturing false positives. The results from the current study aim at a threshold identification
similar to the XSR but differ in methodology. However, the flash flood identification from conditional copula modeling could
be coupled with a module of exposure and a specific vulnerability function. The IDF curve can also provide thresholds for
shorter than 12 h events. For instance, a 6 h long rainfall event with an average intensity above 8.4 mm h$^{-1}$ is potentially flash
flood-inducing. Such an integrated model based on the IDF curves would be an alternative verification to the CCRIF XSR and
reduce model uncertainty.

The methodology proposed here could also be employed for hazard warning services. The Climate Risk Early Warning Sys-
tems (CREWS) Caribbean project aims at strengthening such services. One of the three project components is the institutional
strengthening and capacity building of hydro-meteorological services and early warning systems. The simple decision rule
within the intensity duration space derived in the current study could be adapted for such purposes. More precisely, given a
local weather forecast for the next day and corresponding uncertainty, the risk of a potential flash flood event can be deduced.
After the initial parameterization, a direct implementation into the forecasting routine comes at virtually no cost. Again, even
if there exist other systems, introducing another model based on a different methodology can greatly reduce model uncertainty.

It must be pointed out that the proposed methodology suffers from some shortcomings. The focus on rainfall events as
measured at a certain location ignores general meteorological conditions as well as conditions on the ground. Additional
information such as antecedent rainfall and soil moisture, soil type, or slope gradients can be employed to get a more precise
decision rule. Likely, these factors play a crucial role in the actual development of a hazard given a specific rainfall event.
The current methodology with a bivariate copula at its core is not directly suited for additional variables. While trivariate and
higher dimensional copulas do exist, they are much less understood. Trivariate copulas also impose some limits on the attainable
negative dependence. Furthermore, adding another variable to the copula requires a disproportionately larger sample, where
the sample density decreases exponentially with the number of dimensions. One should note that several of these shortcomings
such as the sample density apply to other methodologies as well. Another potentially more fruitful route might be to consider
separate copula functions for different topography classes or meteorological conditions instead of a unified model that explicitly
accounts for these interdependencies.

The procedure also omits the role of tropical cyclones (TCs). It has long been recognized that in the Caribbean many
instances of extreme rainfall and consequential flooding are due to TCs (Laing, 2004). Ideally, a classification scheme would
take into account synoptic scale weather events. Suppose the proposed classification scheme for flash flood incidents will be

used to estimate the effect of extreme rainfall on the economy or for insurance schemes. In that case, evidence is necessary to distinguish it from TCs (Czajkowski et al., 2017). While the current study did verify via newspaper articles that the flash flood incidents are largely non-TC events, additional care is necessary for applications. For instance, Collalti and Strobl (2022) studies the economic impacts of flooding during tropical storms in Jamaica and finds that only a minor number of heavy rainfall events occur during tropical storms of hurricane strength compared to the number of flash flood incidents discussed in this study.

Still, the analysis has to be geographically interpreted in terms of the whole Island of Jamaica since the cell-wise annual maxima are likely not independent of each other due to weather systems such as TCs affecting more than one cell. This type of dependence due to the simultaneous occurrences of extreme events at multiple stations is also known as spatial coherence or storm dependence and can lead to an underestimation of the risks associated with extreme precipitation at each specific location (Zhang et al., 2022). However, it is particularly challenging to model such dependence adequately in a statistical analysis. This is the main reason why we refrain from making statements concerning a single location which would require the explicit modeling of dependence, for instance, via single-site conditioning (Wadsworth and Tawn, 2022). In other words, by not exploiting the timing or location relative to other observations in the data of yearly maximum events and thus focusing the analysis on the island level, the results are unbiased concerning the spatial dependence of single observations but do not provide any location-specific insights.

Another shortcoming to the generalization of the methodology concerns climate change and the stationarity of the purported relationship. There is a strong consensus that climate change will influence extreme precipitation and consequentially flood risk in many parts of the world, including the Caribbean (IPCC, 2012, 2023). One possibility would be to incorporate non-stationarity in time to the marginal extreme value distributions (Sam et al., 2023). This could be extended to the copula function, as Yin et al. (2018) demonstrates. However, this approach is not feasible in the current application due to the, in climatic terms, short time series of the data. Most climatic variation in a 20-year time series is likely internal climate oscillation like ENSO (Bedoya-Soto et al., 2019; Cai et al., 2020). There could, however, be an alternative route to tackle climate change in settings where the data length is short in the form of (external) climate change allowance (Kay et al., 2021). This would entail re-sampling and adjusting the satellite precipitation data by a climate change allowance factor derived from other studies. Running the methodology of conditional copula modeling with this new data, one could explicitly model the effect of climate change on flood risk and classification in a comparison.

## 6 Conclusions

This study uses 93 confirmed flash flood events in Jamaica from 2001 to 2018 to estimate intensity-duration-frequency (IDF) curves via conditional copula sampling. Rainfall information of flash flood events is taken from remote sensing and additional data on location-specific yearly maximum rainfall events was constructed. This considerably larger sample of statistically similar events allows for higher robustness in the estimation. It further enables one to find an IDF curve threshold above which flash flood events become likely. This threshold corresponds to a return period of $2\frac{1}{6}$ years. A comparison with IDF curves for

Jamaica in Burgess et al. (2015) yields similar results in terms of absolute level, but these are less convex concerning extremely intense or long events. The simple nature of connecting the copula method for IDF curves with a classification for flash floods potentially opens up many applications in parametric insurance programs, regional risk mapping, and hazard warning systems.
The current study abstracts from event-determining conditions other than local rainfall intensity and duration. Future research should, therefore, include other factors, such as soil type and terrain ruggedness in the conditional copula modeling, as well as incorporate synoptic scale meteorological conditions and climate change scenarios.

*Code and data availability.* Shapefiles of all flood events in the study region and period are available from the Jamaican Office of Disaster and Preparedness Management (ODPEM). The Integrated Multi-satellitE Retrievals for GPM (IMERG) data is freely available from NASA.
Cleaned flash flood event data derived from the OPDEM shapefiles are available under https://doi.org/10.48620/364 together with code that merges it with the GPM (IMERG), performs the main analysis and figures.

## Appendix A

### A1    Event Definition

Depending on the IETD, the statistical properties of the events change. Values of IETD between 4h and 24h are considered.
Figure A1 shows how the average intensity and duration changes for the flash flood event data with different values of the IETD. There is a relatively sudden drop in mean intensity for IETDs above 21 hours. Also, the mean duration increases one to one up until an IETD of 12 hours after which the slope becomes flatter. Both indicate that the IETD above 12 and 21 hours results in imprecisely delimited events with regard to duration and intensity, respectively. Figure A2 shows the probability density function of duration and intensity for the confirmed flash flood event data (green) and maximum rainfall events (blue).
In order to apply the two-sample approach, they should follow the same marginal distribution. The maximum rainfall events are more intense but shorter compared to the confirmed flash flood events. The resemblance between duration increases with higher IETD while there is no clear pattern for intensity. In conclusion, an IETD of 12 hours is best suitable for the data at hand. Note that in section 4.3, where marginals are estimated for the conditional copula modeling, Kolmogorov-Smirnov tests indicate that the marginal distributions of the maximum rainfall event data are suitable for the smaller confirmed events data as
well.

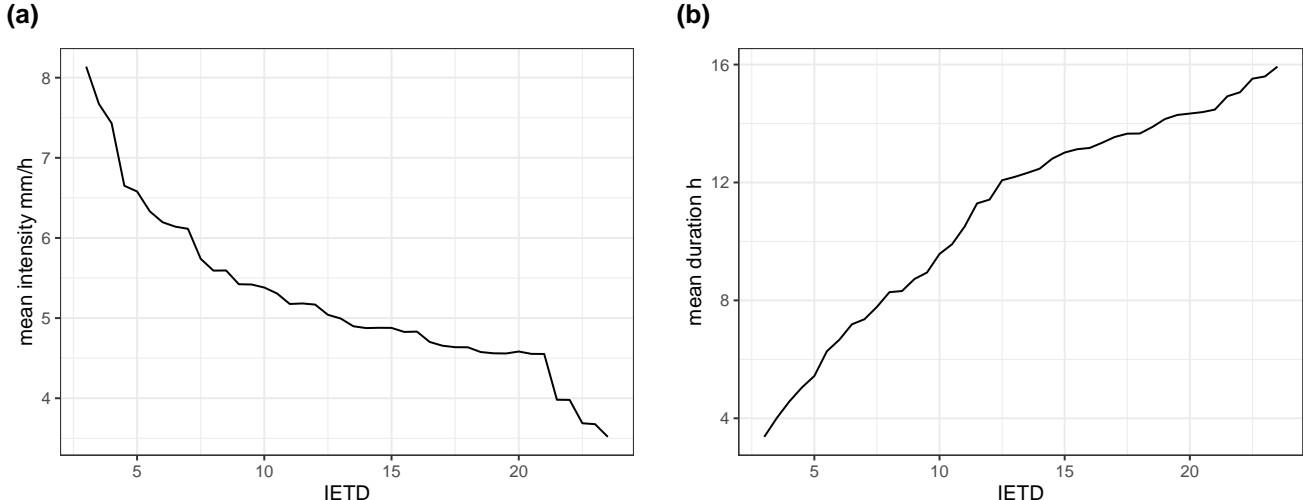

**Figure A1.** Mean event intensity and duration for different values of IETD.

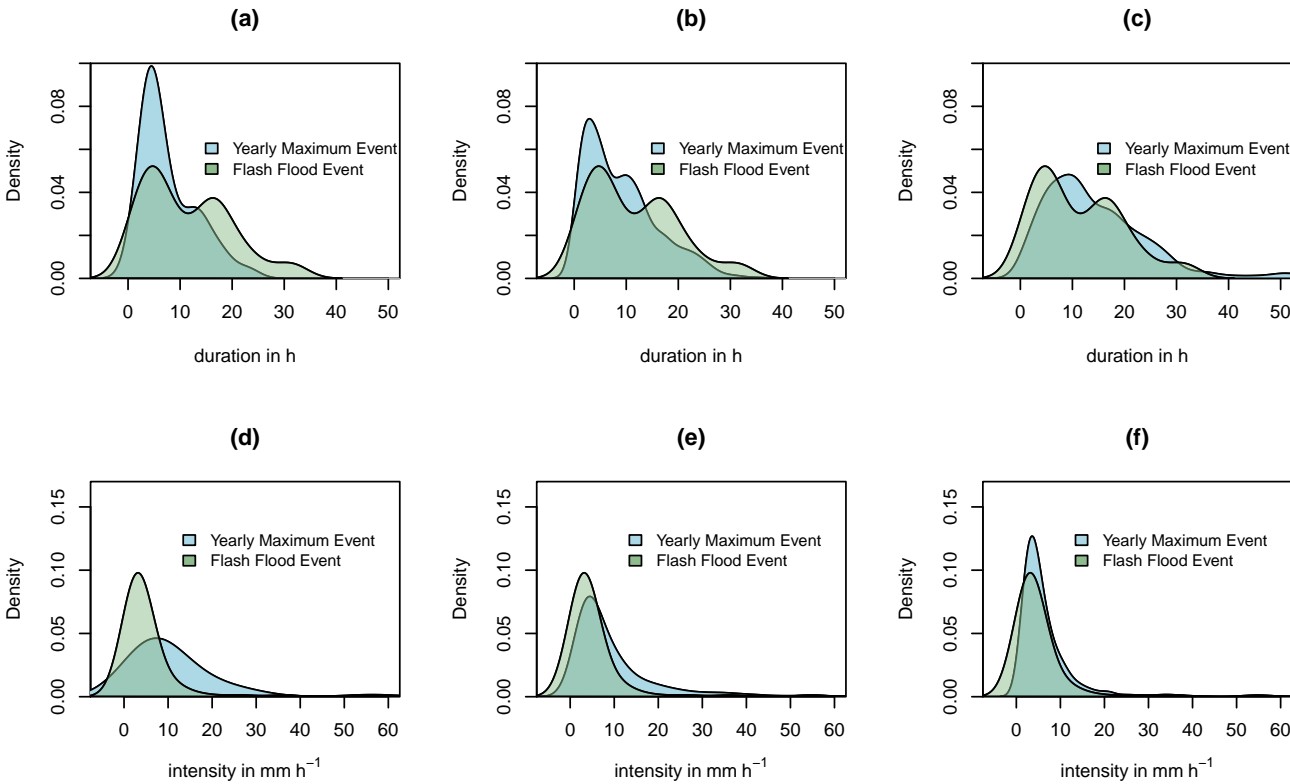

**Figure A2.** Probability density plots for duration and intensity for an IETD of 6 h, 12h, and 24 h.

**Table A1.** Summary Statistics

| | | Maximum Yearly Events | | | |
| --- | --- | --- | --- | --- | --- |
| | N | Mean | St. Dev. | Min | Max |
| Total Rainfall | 1,120 | 47.542 | 24.522 | 6.705 | 148.485 |
| Event Duration | 1,120 | 9.031 | 6.811 | 0.500 | 36.000 |
| Rainfall Intensity | 1,120 | 9.927 | 12.089 | 0.905 | 120.000 |
| Total Rainfall | 93 | 73.677 | 53.283 | 12.120 | 240.100 |
| Event Duration | 93 | 11.419 | 8.325 | 17.000 | 32.000 |
| Rainfall Intensity | 93 | 5.169 | 7.271 | 0.962 | 54.760 |

Summary table of events with IETD of 12 hours for all locations with a confirmed flash flood event.

*Author contributions.* ES and DC conceived the research framework and developed the methodology. DC was responsible for the code compilation, data analysis, graphic visualization, and first draft writing. NS and ES participated in the data collection of this study. All authors discussed the results and contributed to the final version of the paper.

*Competing interests.* We declare no competing interests. This research did not receive any specific grant from funding agencies in the public, commercial, or not-for-profit sectors.

*Acknowledgements.* We would like to thank the participants of the Workshop on Compound Weather and Climate Events 2021 (Bern, Online), and the IPWSD 2021 (Columbia University, Online) for their valuable comments.

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
