# Peer review of "Flash Flood Detection via Copula-based IDF Curves: Evidence from Jamaica"

_Natural Hazards and Earth System Sciences, 2023_

## Author Response (AR1)

Editor Comments:

I think that the Authors correctly answered to all the comments of the referee and accordingly they can improve the paper. Nevertheless, I noticed that all the quoted references are old or very old: I think that this can be a demerit point. Then, I ask to the Authors to quote more recent literature in order to avoid that the reader can think that this sector of the research stopped 20 or 30 years ago.

***Answer:*** Thank you very much for your consideration. We agree that some older references in the original submission can make the manuscript seem outdated. Accordingly, we substituted some of these references with more recent literature and added references, in general.

Reviewer 1 Comments:

1. Introductory comments on the manuscript:

The content of the manuscript is based on theoretical versus empirical probabilistic principles, which inexperienced postgraduate students will find it difficult to assimilate. It is more of abstract mathematical modelling than practical mathematical modelling. This is somewhat a diversion from parametric and nonparametric statistics on extreme value flood predictions. It stands to be established– the comparativeness of the outcome of copula mathematical modelling with those of parametric and nonparametric statistical approach (See Nwaogazie & Sam, 2019).

***Answer:*** We agree with the assessment that the manuscript is based on theoretical principles that derive from mathematical modelling. It is also criticized that the motivating principles do not necessarily relate to parametric and nonparametric modelling in extreme value flood predictions. To address this, we added a paragraph each to the Introduction (starting on line 34, line 37 in markup) and Discussion (line 319, 324 in markup) to make such a comparison explicit and discuss its implications. Further, we added a flow-chart to the Methodology Section (Figure 2) that summarizes the modelling steps and data inputs, so that these can be easily followed by anybody who wants to implement a similar approach. In addition, we also extended the description of conditional copula sampling implementation in Section 4.4. We hope this helps clarify the modelling choices to a broader audience and make it a more readily accessible manuscript.

1. The statement in Section 3.3 "Conditional sampling from the copula enables the construction of IDF curves with T year return periods". The Authors were not very explicit in giving illustrative example or some guideline on how the conditional sampling were carried out for the construction of IDF models. The construction of IDF model is the heart of this research work.

***Answer:*** The conditional copula sampling is not explicitly discussed as it is not a novel contribution of the manuscript. The theoretical basis for the process is the inverse Rosenblatt transformation, also known as the conditional distribution method. However, our exposition could indeed benefit from a clearer exposition. We reworked and extended Section 4.4, "Conditional Copula IDF Curves", to give an illustration of how we implement the procedure. For replication, the code and data repository are available at https://doi.org/10.48620/364.

2.  Section 3.3 on "Two Sample Approach". Basically, the Authors explained the theoretical approach, particularly with respect to IDF curves versus return period, without illustrative examples to drive the point home.

*Answer:* We agree that an example would help to illustrate the proposed methodology. However, we think such an applied example might be better shown in the new Results section 4.6 "Uncertainty". We added Figure 10, which illustrates the classification ratio achieved with different return periods and illustrates how the 2 1/6 years return period IDF curve separates and classifies the maximum rainfall events. Both are discussed in Section 4.6.

3.  Interestingly, the predicted/plotted IDF curves of the copula modelling is in agreement or takes similar features as those predicted using parametric and nonparametric statistical (extreme value approach). What is explained is the duration of 11.4 hours as against 9 hours observed. Also, the IDF curves were limited between 2 & 40 years as against upwards of 200years as reported in literature. This is not reasonably explained as to the shortfall.

*Answer:* We agree that the maximum of a 40-year return period, which is only twice as long as the observational period, is one of the shortfalls of our methodology that should be discussed. While this relatively short return period is of little concern to flash floods that frequently occur, it puts the methodology's potential applicability into perspective for the extreme value approach and its use of other extreme events with much higher return periods. We discuss the trade-off between longer data availability versus spatial resolution in section 4.4 (line 269, 277 in markup). We further reworked this discussion paragraph for this comparison accordingly (line 305, line 313 in markup).

4.  The explanation given as to low predictive capacity of the copula modelling is not satisfactory in the Discussion section as with those developed using long term series data by (Burgess et al., 2015).

*Answer:* We thank the referee for pointing this out and agree that the predictive capacity should be more thoroughly discussed concerning the long time series data by Burgess et al. (2015). We did so in the first paragraph of the discussion (line 305, line 313 in markup).

5.  What happens if we include the effect of climate change in IDF modelling, where it is expected that nonstationary IDF models should be higher than the stationary equivalent all things being equal. See Sam et al. (2023).

*Answer:* The referee's point that climate change influences IDF curves and should therefore be represented in its modelling is well received. This could potentially be done by considering non-stationary extensions of the methodology. However, we fear that the short time period of around 20 years for which both confirmed flash flood events and satellite rainfall data are available makes inference concerning a climate signal extremely speculative in the given setting. We did discuss this issue more thoroughly in a new paragraph in the discussion (line 375, 384 in markup).

Reviewer 2 Comments:

The manuscript "Flash Flood Detection via Copula-based IDF Curves: Evidence from Jamaica" provides an alternative approach for the estimation of design rainfall in terms of IDF values, as well as an approach for forecasters to define an issue-level for flash floods based on rainfall. The manuscript, although quite theoretical, is well written and easy to follow for the reader. The mathematical theory is well founded, however, I have a few suggestions to clarify and elaborate on some climatological aspects.

1. You write that yearly maxima from 1120 grid cells are used in the calculations. Can these annual maxima be considered independent? If not, what implications might it have? Please discuss.

*Answer:* The point raised that the annual maxima from the grid cells can likely not be considered independent is relevant for the analysis. Generally, this type of dependence due to the simultaneous occurrences of extreme events at multiple stations is also known as spatial coherence or storm dependence and can lead to an underestimation of the risks associated with extreme precipitation at each specific location (Zhang et al. (2022)). However, it is particularly challenging to model such dependence appropriately for statistical analysis. This is one of the reasons why we refrain from making statements concerning a single location in our methodology, which would require the modelling of dependence, for instance, via single-site conditioning (Wadsworth et al. (2022)). In other words, by not exploiting the timing or location relative to other observations in the data of yearly maximum events and thus focusing the analysis on the island level, the results are unbiased concerning the spatial dependence of single observations. We discussed this issue in a new paragraph in the discussion (line 365, 374 in markup).

2. Chapter 2.1 describes the study region, but says very little about the rainfall climate in the area, except " the eastern Blue Mountains historically experience the most rainfall". You show some maps of yearly maxima (12 hour) events in Figure 2 but It would be nice to see a climatological map of rainfall and a description of any spatial variability. Maybe a description of rainfall at different durations, and which types of weather systems dominate at the different scales.

*Answer:* This is indeed lacking in the manuscript, and we added subsection 2.2, "Rainfall Climate". We also added Figure 1, showing maps of Jamaica's rainfall climate. This includes the annual rainfall over the sample period as we as the average rainfall distribution for long (72 h IETD) to short (6 h IETD) rainfall events.

3. Same as above, but concerning IDF estimates. You seem to provide one single IDF curve for the whole of Jamaica, or did I misunderstand? Although this might be real, and that the spatial variability is negligible, please discuss it in the text.

*Answer:* We do provide a single IDF curve for Jamaica, mainly for the reasons discussed above. In other words, the analysis abstracts from spatial dependence of rainfall extremes and flood events across Jamaica because of the difficulties involved for a regionalization with constrained data. The IDF curves instead provide a representative estimate for the whole country, on average. This choice and its consequences more clearly explained in the new second to last paragraph of the discussion (line 365, 374 in markup).

4. You present a deterministic estimation of IDF. Could uncertainty, i.e. in terms of confidence intervals, be determined from your model?

*Answer:* The deterministic estimation of the IDF curve and implementation for flash flood detection could be supplemented with a measure of uncertainty regarding the model choice

and parameters. However, the model involves several steps, and it is not clear how the uncertainty of a single step adds up. For instance, while the uncertainty in copula parameter estimation could be employed for uncertainty in the IDF curve, relating this with the uncertainty from marginal distribution estimation is difficult. A consistent approach to consider parameter uncertainty in different steps of the model is to bootstrap confidence intervals. This would still be under the premise that the choice of the copula family and marginal distribution function is subject to uncertainty that cannot be quantified. We did this and added section 4.6 "Uncertainty", where we discuss the results. The new Figure 9 shows the uncertainty from copula and marginal estimation for 2-, 10-, 20- and 40-year return period IDF curves. The new Figure 10 then shows the uncertainty from bootstrapping for the classification.

5. One small language correction on line 24: "warning system typically uses" --> "warning systems typically use"

*Answer:* thank you for pointing out this mistake; we have corrected it.

6. My last comment related to climate change and trends, which your stationary model does not take into account. I see another reviewer already pointed this out. For design and long-term planning purposes future climate change poses a challenge - can you please discuss how this can be tackled, for instance through climate change allowances?

*Answer:* Climate change will undoubtedly influence the flood risk and should therefore be considered when developing a method for detection. In response to the first reviewer, we discussed the possibility of extending the model to be non-stationary. This would be difficult to justify in the current setting with only 20 years of rainfall data. However, the use of climate change rainfall allowance could be a fruitful route to discover the effect of climate change and trends in our setting that we discussed in a new paragraph to the discussion (line 375, 384 in markup).